# Genetic polymorphisms associated with susceptibility to COVID-19 disease and severity: A systematic review and meta-analysis

Cristine Dieter[1,2☉], Letícia de Almeida Brondani[1,2☉], Cristiane Bauermann Leitão[1,2], Fernando Gerchman[1,2], Natália Emerim Lemos[1,2], Daisy Crispim[1,2]*

1 Endocrine and Metabolism Division, Hospital de Clínicas de Porto Alegre, Porto Alegre, Rio Grande do Sul, Brazil, 2 Postgraduate Program in Medical Sciences: Endocrinology, Faculdade de Medicina, Universidade Federal do Rio Grande do Sul, Porto Alegre, Rio Grande do Sul, Brazil

☉ These authors contributed equally to this work.
* dcmoreira@hcpa.edu.br

**Data Availability Statement:** All relevant data are within the manuscript and its Supporting Information files.

## Abstract

Although advanced age and presence of comorbidities significantly impact the variation observed in the clinical symptoms of COVID-19, it has been suggested that genetic variants may also be involved in the disease. Thus, the aim of this study was to perform a systematic review with meta-analysis of the literature to identify genetic polymorphisms that are likely to contribute to COVID-19 pathogenesis. Pubmed, Embase and GWAS Catalog repositories were systematically searched to retrieve articles that investigated associations between polymorphisms and COVID-19. For polymorphisms analyzed in 3 or more studies, pooled OR with 95% CI were calculated using random or fixed effect models in the Stata Software. Sixty-four eligible articles were included in this review. In total, 8 polymorphisms in 7 candidate genes and 74 alleles of the *HLA* loci were analyzed in 3 or more studies. The *HLA-A\*30* and *CCR5* rs333Del alleles were associated with protection against COVID-19 infection, while the *APOE* rs429358C allele was associated with risk for this disease. Regarding COVID-19 severity, the *HLA-A\*33*, *ACE1* Ins, and *TMPRSS2* rs12329760T alleles were associated with protection against severe forms, while the *HLA-B\*38*, *HLA-C\*6*, and *ApoE* rs429358C alleles were associated with risk for severe forms of COVID-19. In conclusion, polymorphisms in the *ApoE*, *ACE1*, *TMPRSS2*, *CCR5*, and *HLA* loci appear to be involved in the susceptibility to and/or severity of COVID-19.

## Introduction

Coronavirus disease 2019 (COVID-19), caused by the severe acute respiratory syndrome coronavirus 2 (SARS-CoV-2), was identified in China near the end of 2019, and progressed to a pandemic condition in March 2020, resulting in a major public health problem worldwide due to its social and economic burdens [1]. As of February 1, 2022, COVID-19 affected more than

**Funding:** This study was partially supported by grants from the Conselho Nacional de Desenvolvimento Científico e Tecnológico (CNPq, grant numbers 401610/2020-9 and 425579/2018-2), Fundo de Incentivo à Pesquisa e Eventos (FIPE) at Hospital de Clínicas de Porto Alegre (grant number: 2020-0218), and Coordenação de Aperfeiçoamento de Pessoal de Nível Superior (CAPES). D.C., C.B.L. and N.E.L are recipients of a scholarship from CNPq, while C.D. is a recipient of scholarship from CAPES.

**Competing interests:** The authors have declared that no competing interests exist.

370 million people, and caused more than 5,658,702 deaths (https://www.who.int/publications/m/item/weekly-operational-update-on-covid-19—1-february-2022).

Clinical manifestations of COVID-19 vary from an asymptomatic infection, dry cough, sore throat, fever, shortness of breath, fatigue, muscle pain, headache, loss of taste or smell, vomiting, diarrhea, to acute respiratory distress syndrome. Approximately 15% of patients develop the severe form, which can progress to pneumonia, respiratory failure, kidney injury, multiorgan dysfunction, and death [2, 3]. The variation in symptoms and severity of COVID-19 is partially explained by known risk factors, including advanced age, male gender, and presence of comorbidities, such as diabetes, obesity, hypertension, and heart disease [4, 5]. However, severe outcomes have also been observed in young and healthy patients, suggesting that other risk factors, such as genetic predisposition, may increase the risk to and/or severity of this disease [6–8].

It is well known that host genetic polymorphisms play a key role in the susceptibility or resistance to different viral infections [9, 10]. Taking into account the main role of host genes in the entry and replication of SARS-CoV-2 in cells and in mounting the immune response, it seems that a combination of multiple genes might be involved in COVID-19 pathogenesis [9]. Accordingly, to date, numerous studies have been conducted on the association between genetic polymorphisms and COVID-19 [6, 7, 9–11]. Some studies have indicated that polymorphisms in genes related to innate and adaptive immune response [toll-like receptors (*TLRs*), human leukocyte antigen (*HLA*) class I and II, and cytokines/chemokines] and in genes involved in viral binding and entry into host cells (angiotensin converting enzyme-2 –*ACE2*, and transmembrane serine protease–*TMPRSS*) are associated with COVID-19 development and/or severity [6–8, 12]. However, it is still unclear which and to what degree specific polymorphisms contribute to the susceptibility for this disease [6].

Thus, aiming to identify the genetic factors that may influence COVID-19 susceptibility and severity, we conducted a comprehensive and updated systematic review of the literature on the subject followed by meta-analyses of those polymorphisms analyzed in three or more studies. Even though few systematic reviews have been published regarding the association between polymorphisms in different genes and COVID-19 [6, 7, 10, 12].

## Materials and methods

### Literature search strategy and eligibility criteria

This comprehensive and updated systematic review was performed and written according to the Preferred Reporting Items for Systematic Reviews and Meta-Analyses (PRISMA), Meta-analysis of Observational Studies in Epidemiology (MOOSE) statements and guideline for Systematic Reviews of Genetic Association Studies [13–15], and it was registered at PROSPERO (http://www.crd.york.ac.uk/PROSPERO) under the CRD42021248091 number. We performed a search at PubMed and Embase repositories for all English, Portuguese, and Spanish language original articles that analyzed potential associations between genetic polymorphisms and susceptibility/severity for COVID-19, up to July, 2021. For this, the following MeSH terms were used: (SARS-CoV-2 OR COVID-19 OR severe acute respiratory syndrome OR SARS virus) AND (polymorphism, genetic OR polymorphism, single nucleotide OR polymorphism, single-stranded conformational OR polymorphism, restriction fragment length OR DNA copy number variations OR amplified fragment length polymorphism analysis OR mutation OR mutation rate OR INDEL mutation OR mutation, missense OR point mutation OR frameshift mutation OR codon, nonsense). In addition, studies of interest were also searched in the GWAS Catalog (https://www.ebi.ac.uk/gwas).

Two independent investigators (C.D and L.A.B) screened and evaluated the eligibility of each study retrieved from the online repositories by reviewing titles and abstracts. When abstracts did not provide adequate information, the full texts of the extracted articles were also reviewed, as previously reported by our group [16, 17]. Discrepancies between the two investigators were settled by debate between them and, when necessary, a third reviewer (D.C.) was consulted. All observational human studies that compared frequencies of at least one polymorphism between patients with and without COVID-19 or between COVID-19 patients with different degrees of severity were included in this systematic review. Moreover, reference lists coming from the articles fulfilling our eligibility criteria were manually searched to identify other potentially relevant citations.

The exclusion criteria were: 1) articles without enough data to estimate an OR with 95% CI; 2) duplicated studies (in this case, the most complete study was chosen for inclusion); and 3) non-human studies.

## Data extraction and quality evaluation

Necessary information from each study was individually extracted by C.D. and L.A.B. using a standardized form [16, 17]. Agreement was pursued in all evaluated items of this form; however, when an agreement could not be reached, divergences in data extraction were solved by referring to the original article or by consulting another investigator (D.C.). Data retrieved from each study were as follows: 1) characteristics of the studies and samples (including publication year, name of first author, number of subjects in each analyzed group, mean age, gender, country, and ethnicity); and 2) data of the polymorphisms of interest [including their identification, allele/genotype frequencies, and OR (95% CI)]. When data were not available in the article, the authors were contacted by email for the necessary information, but only part of them answered.

The Clark-Baudouin Score (CBS) was used to evaluate the quality of the included studies [18]. This score applies pre-defined criteria to assess each publication, highlighting quality issues in the conduction of studies and interpretation of results. Using a 10-point scoring sheet, investigators can evaluate sections of the articles related to reproducibility, selection of subjects, statistical analyses, and genotyping methods.

## Statistical analyses for meta-analysis

Those polymorphisms analyzed in three or more studies were submitted to meta-analyses using the Stata 15.0 software (StataCorp, College Station, TX, USA). Goodness-of-fitness $\chi^2$ tests were used to evaluate whether genotype frequencies were in conformity with the Hardy-Weinberg Equilibrium (HWE) in the control groups. Associations between individual polymorphisms and COVID-19 susceptibility and/or severity were analyzed using OR (95% CI) calculations for the allele contrast, dominant, recessive, and additive inheritance models, categorized as suggested by a previous publication [19]. For the *HLA* allelic analysis, frequency was calculated as the number of cases or controls harbouring at least one positive event (one allele type) divided by the total number of chromosomes included in each of the corresponding groups [20]. Inter-studies heterogeneity was tested using $\chi^2$-based Cochran's Q statistic, while inconsistency was quantified with the $I^2$ metric [21, 22]. When $P < 0.10$ (Q statistic) and/or $I^2 > 50\%$, heterogeneity was considered statistically relevant. In this case, the DerSimonian and Laird random effect model (REM) was used to calculate OR (95% CI) for each study and for the pooled effect. In the lack of significant inter-studies heterogeneity, the fixed effect model (FEM) was used for this calculation.

# Results

## Literature search

Fig 1 shows the flow diagram illustrating the strategy used to identify and select studies for inclusion in our systematic review and meta-analyses. A total of 2936 articles were retrieved after searching PubMed, Embase, and GWAS Catalog resources, and 2727 of them were excluded during the review of titles and abstracts due to disagreements with our defined eligibility criteria. Two hundred and nine articles remained to be full text evaluation. Nevertheless, after carefully analyzing the full texts, another 145 studies were excluded, and a total of 64 articles were included in this systematic review (Table 1 and Fig 1). Among them, 30 studies, where the same SNP was evaluated in at least 3 articles and frequency data was available, were included in the meta-analyses.

## Qualitative synthesis of studies that analyzed associations of SNPs and COVID-19

Table 1 shows the compiled main data of the 64 eligible studies included in this systematic review. More than 200 polymorphisms and 50 genes/loci were studied regarding their

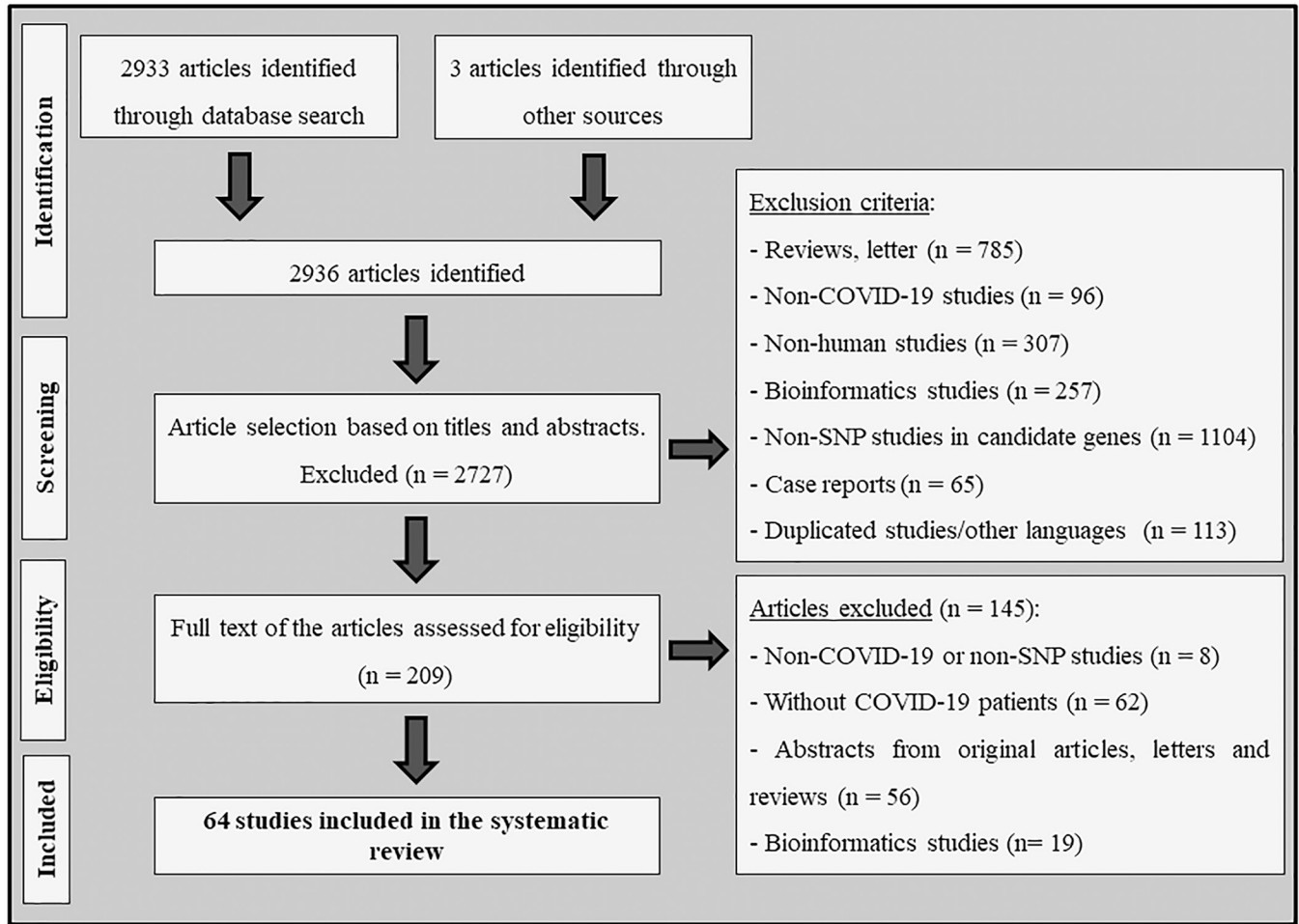

**Fig 1. Flowchart illustrating the search strategy used to identify association studies between genetic polymorphisms and COVID-19 disease.**

**Table 1. Characteristics of studies included in the systematic review.**

| Reference | Population | Sample (case/control) | Gene | Results |
|---|---|---|---|---|
| Agwa et al., 2021 [23] | Egyptian | 141 cases / 100 controls | INFλ, TLL1, DDR1 | Disease susceptibility: The IFN-λ rs12979860 C/C, TLL1 rs17047200 A/A and the DDR1 rs4618569 A/A genotypes were associated with COVID-19 (P = 0.011, P = 0.012, and P = 0.026, respectively). Severity: The DDR1 rs4618569 A/G was associated with COVID-19 severity (P = 0.007). |
| Alghamdi et al., 2021 [24] | Saudi | 880 cases | IFITM3 | Disease susceptibility: The rs12252 G allele was associated with risk for hospital admission (OR = 1.65, 95% CI 1.01–2.70, P = 0.04). Severity: The rs12252 G allele conferred risk for mortality (OR = 2.2, 95% CI 1.16–4.20, P = 0.01). |
| Amodio et al., 2020 [25] | Italian | 381 cases | IFNL3, IFNL4 | Severity: The IFNL4 rs368234815 DelG/DelG genotype was associated with risk for higher viral loads in COVID-19 patients (OR = 1.24, 95% CI 1.09–1.40). |
| Amoroso et al., 2021 [26] | Italian | 219 cases /40,685 controls | HLA-A, -B, -DRB1 | Disease susceptibility: The HLA-DRB1*08 allele was associated with risk for COVID-19 (OR = 1.9, 95% CI 1.2–3.1, P = 0.003) Severity: The HLA-DRB1*08 allele conferred risk for death (OR = 2.9, 95% CI 1.15–7.21, P = 0.023). |
| Avendaño-Félix et al., 2021 [27] | Mexican | 193 cases | IL-10 | Severity: The rs1800871 and rs1800872 polymorphisms were not associated with COVID-19 severity (P = 0.286 and P = 0.235, respectively) and related-outcomes (P = 0.499 and P = 0.531). |
| Benetti et al., 2020 [28] | Italian | 131 cases /258 controls | WES | Disease susceptibility: ACE2 allelic variability was higher in control group compared to the patient cohort, detected from a cumulative analysis of the identified variants (P <0.029). |
| Benetti et al., 2020 [29] | Italian | 35 cases / 150 controls | WES | Disease susceptibility: Through the gene burden test, mutations in PRKRA and LAPTM4B genes were identified as being risk factors, while mutations in OR4C5 and NDU-FAF7 genes represented protective factors for COVID-19. |
| Bernas et al., 2021 [30] | German | 4758 cases /10,5008 controls | CCR5 | Disease susceptibility: The CCR5 Δ32 polymorphism was not associated with COVID-19 (OR = 0.96, 95% CI 0.89–1.03, P = 0.21). Severity: The CCR5 Δ32 polymorphism did not differ significantly between individuals with or without symptomatic infection (OR = 1.13, 95% CI 0.88–1.45, P = 0.32), severe respiratory tract infection (OR = 1.03, 95% CI 0.88–1.22, P = 0.68) or respiratory hospitalization (OR = 1.16, 95% CI 0.79–1.69, P = 0.45). |
| Cabrera-Marante et al., 2020 [31] | Latin-american, Spanish, Polish | 22 cases | PRF1 | Severity: Two of 22 patients showed PRF1 A91V mutation in heterozygosis (allele frequency = 0.045). These 2 A91V-positive patients had higher fever associated with respiratory symptoms and died. |
| Cafiero et al., 2021 [32] | Italian | 104 cases | ACE1, ACE2, AGT, AGTR1 | Severity: The ACE2 rs2074192 T, ACE1 Del, and AGT rs699 C alleles were more frequent in symptomatic patients vs. asymptomatic (P = 0.001, P <0.001, and P = 0.033, respectively). |
| Calabrese et al., 2020 [33] | Italian | 68 cases / 222 controls | ACE1 | Severity: The frequency of ACE1 Del/Del genotype was higher in COVID-19 patients with pulmonary embolism (PE) than patients without PE (72 vs. 46.5%; P = 0.048). |

(*Continued*)

**Table 1.** (*Continued*)

| Reference | Population | Sample (case/control) | Gene | Results |
|---|---|---|---|---|
| Cantalupo *et al.*, 2021 [34] | Italy | 202 cases /929 controls (rs35951367) 221 cases/1084 controls (rs3441865) 147 cases / 1095 controls (rs333) | WES | Disease susceptibility: The *CCR5* rs35951367 C allele was associated with risk for COVID-19 (OR = 1.307, 95% CI 1.01–1.70, P = 0.043). The *CCR5* rs34418657 G/T genotype was more frequent in patients with COVID-19 than controls (OR = 3.978, 95% CI 1.060–14.933, P = 0.027). No association was found between the *CCR5* Δ32 (rs333) polymorphism and COVID-19 (P = 0.99). |
| Coto *et al.*, 2021 [35] | Spanish | 318 cases / 350 controls | ABO | Disease susceptibility: The rs8176719 polymorphism was not associated with risk for COVID-19 or disease severity. |
| Cuesta-Llavona *et al.*, 2021 [36] | Spanish | 801 cases / 650 controls | CCR5 | Disease susceptibility: Homozygosis for the *CCR5* Δ32 deletion (rs333) conferred protection against COVID-19 (OR = 0.66, 95% CI 0.49–0.88, P = 0.01). |
| Del Ser *et al.*, 2021 [37] | Spanish | 62 cases / 851 controls | APOE | Disease susceptibility: The *APOE* ε4 allele was associated with the presence of symptoms of COVID-19 (OR = 1.85, 95% CI 1.13–2.88, P = 0.010). |
| Dite *et al.*, 2021 [38] | British | 1582 cases[a] | Array | Severity: A score of 64 SNPs was associated with risk for COVID-19 severity (OR = 1.19, 95% CI 1.15–1.22, P <0.001). A model incorporating this score and clinical risk factors showed 111% better discrimination of disease severity than a model with just age and gender. |
| Ellinghaus *et al.*, 2020 [39] | Italian, Spanish | 835 cases / 1255 controls 775 cases/ 950 controls | GSA | Severity: The 3p21.31 cluster was identified as a susceptibility locus in patients with COVID-19 with respiratory failure (OR = 1.77, 95% CI 1.48–2.11; P = $1.15 \times 10^{-10}$). |
| Gavriilaki *et al.*, 2021 [40] | Greek | 97 cases | NGS | Severity: Patients carrying the *THBD* rs1042580 C and *CFH* rs800292 G alleles did not require ICU hospitalization (*vs.* patients carrying the other alleles). Polymorphisms in *ADAMTS13, C3* and *CFH* genes were associated with risk for ICU hospitalization (P = 0.022). |
| Gómez *et al.*, 2020 [41] | Spanish | 204 cases / 536 controls | ACE1, ACE2 | Severity: The *ACE1* Del/Del genotype was associated with severe COVID-19 (P = 0.049). The *ACE2* rs2285666 polymorphism was not associated with disease severity. |
| Gómez *et al.*, 2021 [42] | Spanish | 311 cases / 440 controls | IFITM3 | Disease susceptibility: The *IFITM3* rs12252 C allele was associated with risk for COVID-19 hospitalization after adjustment by age and gender (OR = 2.02, 95%CI 1.19–3.42, P = 0.01). |
| Grimaudo *et al.*, 2021 [43] | Italian | 383 cases | MERTK, INFL4, PNPLA3, TLL1 | Severity: In patients younger than 65 years, the *PNPLA3* rs738409 G/G (OR = 4.69, 95% CI 1.01–22.04, P = 0.049) and *TLL1* rs17047200 T/T (OR = 9.1, 95% CI 1.45–57.3, P = 0.018) genotypes were associated with risk for disease severity. |
| Gunal *et al.*, 2021 [44] | Turkish | 90 cases | ACE1 | Severity: The *ACE1* Ins/Ins genotype conferred protection against severe COVID-19 (OR = 0.323, 95% CI 0.112–0.929, P = 0.036). |
| Hamet *et al.*, 2021 [45] | British | 1644 cases / 15962 controls[a] | Array | Severity: The *ACE2* rs2074192 T allele was associated with more severe outcomes of COVID-19 in obese smoking males of 50 years or older (OR = 4.07, P = 0.036). |
| Hubacek *et al.*, 2021 [46] | Czech | 416 cases / 2404 controls[d] | CCR5 | Severity: The frequency of CCR5 Δ32 allele was higher in COVID-19 asymptomatic patients (23.8%) than COVID-19-symptomatic patients (16.7%) (P = 0.03). |
| Hubacek *et al.*, 2021 [47] | Czech | 408 cases / 2559 controls[d] | ACE1 | Disease susceptibility: The frequency of *ACE1* Ins/Ins genotype was higher in COVID-19 patients *vs.* controls (26.2% *vs.* 21.2%; OR = 1.55, 95% CI 1.17–2.05, P = 0.02). |

(*Continued*)

**Table 1.** (*Continued*)

| Reference | Population | Sample (case/control) | Gene | Results |
|---|---|---|---|---|
| Hubacek *et al.*, 2021 [46] | Czech | 408 cases / 2606 controls[d] | *APOE* | Disease susceptibility: The frequency of the APOE4 allele did not differ between the group of SARS-CoV-2-positive subjects and the control population (P = 0.11). Severity: The presence of least one *APOE4* allele was higher in symptomatic COVID-19 subjects than controls (OR = 1.43, 95% CI 1.05–1.95, P = 0.03). Genotype frequencies were almost identical in COVID-19-asymptomatic subjects and in the control group population (P = 0.86). |
| Karakas Çelik *et al.*, 2021 [48] | Turkish | 155 cases | *ACE1, ACE2* | Severity: *ACE1* Ins/Del and *ACE2* rs2106809 and rs2285666 polymorphisms were not associated with COVID-19 severity. |
| Kerget *et al.*, 2021 [49] | Turkish | 70 cases | *IL-6* | Severity: The *IL-6* rs2074192 G/G genotype was associated with COVID-19 severity (P = 0.002). |
| Kolin *et al.*, 2020 [50] | British | 968 cases / 1734 controls[a] | *Array* | Disease susceptibility: Genome-wide association analysis did not show any significant loci in the meta-analysis (P >0.050). |
| Kuo *et al.*, 2020 [51] | British | 622 cases / 322326 controls[a] | *Array* | Disease susceptibility: The *ApoE* ε4ε4 genotype was associated with risk of COVID-19 positivity (OR = 2.24, 95% CI 1.72–2.93, P = $3.24 \times 10^{-9}$) *vs.* e3e3 genotype. Severity: The presence of the *ApoE* ε4ε4 genotype conferred risk for mortality (OR = 4.29, 95% CI 2.38–7.72, P = $1.22 \times 10^{-6}$) *vs.* e3e3 genotype. |
| Latini *et al.*, 2020 [52] | Italian | 131 cases / Controls[e] | *WES* | Disease susceptibility: *Furin* rs769208985 A and *TMPRSS2* rs114363287 A alleles were more frequent in COVID-19 than GnomAD controls (P = 0.005 and P = 0.016, respectively). *TMPRSS2* rs75603675 T and rs12329760 A alleles were less frequent in COVID-19 patients than GnomAD (P = 0.0446 and P = 0.023, respectively). |
| Lehrer *et al.*, 2021 [53] | British | 688 cases[a] | *S1R* | Severity: The *S1R* rs17775810 T/T genotype was associated with the lowest death rate (0%, P = 0.020). |
| Lehrer *et al.*, 2021 [54] | British | 712 cases / 9265 controls[a] | GWAS-Chr9 | Disease susceptibility: No association was found between the rs657252 polymorphism in Chr9 and COVID-19. |
| Littera *et al.*, 2020 [55] | Italian | 182 cases / 619 controls | *HLA-A, -B, -C, -DRB1* | Disease susceptibility: The haplotype *HLA-A*02:05, B*58:01, C*07:01, DRB1*03:01* protected against SARS-CoV-2 infection. *HLA-C*04:01* allele and the haplotype *HLA-A*30:02, B*14:02, C*08:02* (OR = 3.8, 95% CI 1.8–8.1, P = 0.025) were more frequent in patients than controls. Severity: *HLA-DRB1*08:01* allele was only present in hospitalized patients (OR >2.5, 95% CI 2.7–220.6, P = 0.024). |
| Lorente *et al.*, 2020 [56] | Spanish | 72 cases / 3,886 controls | *HLA-A, -B, -C, -DRB1, -DQB1* | Severity: The *HLA-A*11, HLA-C*01* and *HLA-DQB1*04* alleles were associated with higher mortality due to COVID-19 (OR = 7.69, 95% CI 1.06–55.65, P = 0.040; OR = 11.18, 95% CI 1.05–118.70, P = 0.040; and OR = 9.96, 95% CI 1.23–80.36, P = 0.030; respectively). |
| Malaquias *et al.*, 2020 [57] | Brazilian | 6 cases / 11 controls | *MBL2* | Disease susceptibility: The rs180040 A/A, rs1800451 G/G and rs5030737 C/C genotypes had a higher prevalence in the COVID-19 group. |
| Martínez-Sanz *et al.*, 2021 [58] | Spanish | 39 cases / 28 controls | *Array* | Disease susceptibility: The *ACE2* rs2106806 A (OR = 3.75, 95% CI 1.23–11.43, P = 0.015) and rs6629110 T (OR = 3.39, 95% CI 1.09–10.56, P = 0.028) alleles were associated with risk for COVID-19. |

(*Continued*)

**Table 1.** (Continued)

| Reference | Population | Sample (case/control) | Gene | Results |
|---|---|---|---|---|
| Medetalibeyouglu et al., 2021 [59] | Turkish | 284 cases / 100 controls | MBL2 | Disease susceptibility: The B/B genotype of the codon 54 A/B (Gly54Asp: rs1800450) variant in the MBL2 gene was more frequent in COVID-19 cases vs. controls (10.9% vs. 1.0%; OR = 12.1, 95% CI 1.6–90.1, P = 0.001). |
| Möhlendick et al., 2021 [60] | Germany | 297 cases / 253 controls | ACE1, ACE2 | Disease susceptibility: The ACE2 rs2285666 G/G genotype was associated with risk for COVID-19 (OR = 1.91, 95% CI 1.13–3.24, P = 0.02). No association was found between the ACE1 rs1799752 polymorphism and COVID-19.<br>Severity: The ACE2 rs2285666 G/G genotype confer risk for serious course of COVID-19 compared to moderate course (OR = 3.04, 95% CI 1.47–6.27, P = 0.002) and is also associated with mortality (OR = 2.69, 95% CI 1.02–7.11, P = 0.05). |
| Monticelli et al., 2021 [61] | Italian | 1177 cases[b] | WES | Severity: The TMPRSS2 rs2298659 A and the rs12329760 T alleles were more frequent among mild cases of COVID-19 than severe cases (P = 0.004 and P = 0.029, respectively). |
| Naemi et al., 2021 [62] | Asian | 95 cases | HLA-A, -B, -C, -DRB1, -DQA1, -DQB1 | Severity: No association was found between these HLA genotypes and COVID-19 severity. |
| Novelli et al., 2020 [63] | Italian | 131 cases / 1000 Controls[e] | WES | Disease susceptibility: No association was found between ACE2 polymorphisms (rs140312271, rs2285666 and rs41303171) and COVID-19. |
| Novelli et al., 2020 [64] | Italian | 99 cases / 1017 controls | NGS | Disease susceptibility: The frequencies of three HLA alleles were higher in cases vs. controls: HLA B*27:07 (2.02% vs. 0.10%; P = 0.004), DRB1*15:01 (10.10% vs. 4.62%, P = 0.048), and DQB1*06:02 (7.58% vs. 3.64%, P = 0.016). |
| Pairo-Castineira et al., 2021 [65] | | 2244 cases[c] | GWAS | Severity: Polymorphisms in Chr 12q24.13 (rs10735079, P = $1.65 \times 10^{-8}$, near to OAS1, OAS2 and OAS3 genes), Chr 19p13.2 (rs74956615, P = $2.3 \times 10^{-8}$, near TYK2), Chr 19p13.3 (rs2109069, P = $3.98 \times 10^{-12}$, in DPP9), and Chr 21q22.1 (rs2236757, P = $4.99 \times 10^{-8}$, in IFNAR2) were associated with COVID-19 severity. |
| Petrazzuolo et al., 2020 [66] | French | 140 cases | FPR1 | Severity: No association was found between the FPR1 rs5030880 and rs867228 polymorphisms and COVID-19 severity. |
| Posadas-Sánchez et al., 2021 [67] | Mexican | 90 cases / 263 controls | DPP4 | Disease susceptibility: The DPP4 rs3788979 T/T genotype was associated with risk for COVID-19 (OR = 4.28, 95% CI 2.12–8.62, P = $4.7 \times 10^{-5}$; recessive model). |
| Ravikanth et al., 2021 [68] | Indian | 510 cases / 500 controls | WES | Severity: The TMPRSS2 rs12329760 A allele was less frequent in patients with mild-to-moderate (P = 0.004) or severe disease (P = 0.010) vs. asymptomatic patients. |
| Russo et al., 2021 [69] | Italian | 500 cases / 283 controls | WES | Severity: The TNFRSF13 rs61756766 C allele was more frequent in severe cases vs. non-severe (OR = 11.5, 95% CI 1.3–100, P = 0.010) and asymptomatic patients (OR = 3.7, 95% CI 1.3–10.6, P = 0.020). |
| Saleh et al., 2021 [70] | Egyptian | 900 cases / 184 controls | TNFA | Disease susceptibility: The A/A genotype of the TNF G308A polymorphism was associated with risk for COVID-19 (OR = 3.06, 95% CI 1.26–7.44, P = 0.019). |
| Salem Hareedy et al., 2021 [71] | Egyptian | 46 cases / 14 controls | CYP2D6*4, CYP2D6*2XN, CYP3A4*1B, CYP3A5*3 | Disease susceptibility: Carriers of the CYP2D*2XN C/C genotype had the lower risk for a positive anti-COVID-19 IgG or IgM. The CYP3A4*1B A/A genotype conferred protection against positive anti-COVID-19 IgM (vs. G/G genotype). |

(Continued)

**Table 1.** (Continued)

| Reference | Population | Sample (case/control) | Gene | Results |
|---|---|---|---|---|
| Schönfelder *et al.*, 2021 [72] | Germany | 239 cases / 253 controls | *IFITIM3* | Disease susceptibility: The *IFITIM3* rs12252 and rs34481144 polymorphisms were not associated with COVID-19 development (OR = 1.37, 95% CI 0.73–2.58, P = 0.340; OR = 0.96, 95% CI 0.65–1.41, P = 0.840; respectively).<br>Severity: The *IFITIM3* rs12252 and rs34481144 polymorphisms did not confer risk to COVID-19 severity (OR = 0.89, 95% CI 0.35–2.25, P = 1.00; OR = 1.77, 95% CI 0.94–3.32, P = 0.100; respectively). |
| Schönfelder *et al.*, 2021 [73] | Germany | 239 cases / 253 controls | *TMPRSS2* | Disease susceptibility: The *TMPRSS2* rs383510 C/C genotype was associated with risk for COVID-19 infection (OR = 1.73, 95% CI 1.15–2.59, P = 0.010). The rs2070788 and rs12329760 polymorphisms were not associated with COVID-19. |
| Scutt *et al.*, 2021 [74] | British | 705 cases / 471506 controls[a] | *Array* | Disease susceptibility: The *INK4A/ARF* rs10757278 G/A genotype was associated with lower risk of hospital admission for COVID-19 in non-Caucasian patients (A/A + G/G *vs.* A/G; OR = 0.56, 95% CI 0.37–0.85, P = 0.006). |
| Shikov *et al.*, 2020 [75] | Russian | 37 cases /21 controls | *ACE2, ACE1* | Disease susceptibility: No association was found between *ACE2* and *ACE1* polymorphisms and COVID-19. |
| Shkunikov *et al.*, 2021 [76] | Russian | 111 cases / 428 controls | *NGS* | Disease susceptibility: The *HLA-A*$^*$*01:01* allele was associated with risk for COVID-19, while the *HLA-A*$^*$*02:01* and *HLA-A*$^*$*03:01* alleles conferred protection. |
| Torre-Fuentes *et al.*, 2021 [77] | Spanish | 4 cases / 71 controls | *WES* | Disease susceptibility: No association was found between *ACE2, TMPRSS2* and *FURIN* polymorphisms and COVID-19. |
| Valenti *et al.*, 2021 [78] | Spanish | 72 cases | *Chr3* | Severity: The rs11385942 G/A genotype was associated with COVID-19 severity. |
| Verma *et al.*, 2021 [79] | Indian | 269 cases | *ACE1* | Severity: The *ACE1* Del/Del genotype was associated with risk for severe COVID-19 (OR = 3.69, 95% CI 1.612–8.431, P = 0.002). |
| Vietzen *et al.*, 2021 [80] | | 361 cases / 260 controls | *HLA-E, KLRC2* | Disease susceptibility: The *KLRC2* Del allele conferred risk for hospitalization (OR = 2.6, P = 0.0006) and hospitalization in ICU (OR = 7.1, P <0.0001) *vs.* non-hospitalized patients and controls.<br>Severity: The *HLA-E*$^*$*0101* allele was also associated with risk for hospitalization (OR = 2.1, P = 0.010) and hospitalization in ICU (OR = 2.7, P = 0.010). |
| Wang *et al.*, 2020 [81] | Chinese | 332 cases | GWAS$^*$ / *HLA-A, -B, -C, -DRB1, -DQB1, -DPB1, -DQA1* | Severity: The *TMEM189–UBE2V1* rs6020298 A allele was more frequent in patients with severe COVID-19 than non-severe patients (0.59 *vs.* 0.45) and conferred risk for mild + severe disease (OR = 1.2, P = 4.1 x $10^{-6}$). The *TMPRSS2* rs12329760 minor allele was less frequent among patients with severe COVID-19 *vs.* mild symptomatic patients. *HLA-A*$^*$ *11:01, B*$^*$*51:01*, and *C*$^*$*14:02* alleles were associated with risk for severe COVID-19. |
| Wang *et al.*, 2020 [82] | Chinese | 82 cases / 3548 controls | *NGS* | Disease susceptibility: *HLA-B*$^*$*15:27* and HLA-C$^*$*07:29* were associated with risk for COVID-19 disease (OR = 3.59; 95% CI 1.72–7.50, P = 0.030; and OR = 130.20, 95% CI 5.28–3211, P = 0.025, respectively). |
| Wulandari *et al.*, 2021 [83] | Indonesian | 95 cases | *TMPRSS2* | Severity: No association was found between the rs12329760 polymorphism and COVID-19 severity. |

(*Continued*)

**Table 1.** (Continued)

| Reference | Population | Sample (case/control) | Gene | Results |
|---|---|---|---|---|
| Zhang *et al.*, 2020 [84] | China | 80 cases | *IFITM3* | Severity: The *IFITM3* rs12252 C/C genotype was associated with disease severity in an age-dependent manner (OR = 6.37, P <0.001). |
| Zhou *et al.*, 2020 [85] | British | 1091 cases / 2793 controls[a] | *TMPRSS2, ACE2* | Disease susceptibility: After analyzing 17 and 31 tag SNPs of *ACE2* and *TMPRSS2* genes, respectively, the rs7282236 SNP in *TMPRSS2* gene was the only one associated with risk of COVID-19 disease (OR = 1.33, 95% CI 1.14–1.54, P = $2.31 \times 10^{-4}$). |

Chr: chromosome; GSA: Global Screening Array; GWAS: Genome-wide Association Study; ICU: intensive care unit, NGS: next-generation sequencing; WES: Whole exome sequencing

[a]data from UK biobank

[b]data from GEN-COVID Multicenter Study

[c]data from GenOMICC database

[d]controls data from post-MONICA study

[e]controls data from GnomAD database.

associations with COVID-19 susceptibility or severity of this disease. Most of the studies compared polymorphism frequencies in patients who tested positive for COVID-19 compared to negative controls. Twenty-three studies evaluated polymorphisms in COVID-19 patients categorized according to different degrees of disease severity. **S1 Table** shows the quality of all studies included in this systematic review, which was evaluated using the CBS as described in the Methods Section. Considering a score system that ranges from 0 to 10 points according to the adherence to pre-defined criteria, none of the studies reached 9 points. However, the majority of the studies (70.1%) were classified as presenting good quality since they were awarded 6 to 8 points. The remaining articles were awarded with less than 6 points.

More information regarding the COVID-19 diagnostic criteria, definition of severity degrees, age, ethnicity, gender, and genotyping techniques are described in **S2 Table.** The most studied candidate genes/loci were: *HLA*, *ABO*, *ACE1*, *ACE2*, *APOE*, *CCR5*, *TMPRSS2*, and *IFITM3*. In total, 8 polymorphisms in 7 candidate genes and 74 alleles of the *HLA* loci (*A*, *B*, *C*, *DRB1*, *DQA1*, and *DQB1*) were analyzed in ≥3 studies and subsequently included in the meta-analyses.

## Meta-analyses of *ACE2*, *ACE1*, and *TMPRSS2* polymorphisms

Two polymorphisms in the *ACE2* gene were included in meta-analyses (**Table 2**). The pooled data of 3 studies for the rs41303171 (T/C) polymorphism [28, 63, 77] and 3 studies for the rs2285666 (C/T) polymorphism [41, 58, 63] indicated no association between them and the risk for COVID-19.

The rs1799752 (Ins/Del) polymorphism in the *ACE1* gene was analyzed in 4 studies [33, 41, 47, 60] and the meta-analysis indicated no association between the Ins allele and the risk for COVID-19 (**Table 2**). Regarding COVID-19 severity, 8 studies [32, 33, 44, 47, 48, 58, 60, 79] were included. However, we analyzed the pooled data from 5 studies [41, 44, 48, 60, 79] that included severe COVID-19 patients compared to other degrees of severity (moderate, mild and/or asymptomatic). The meta-analysis of these studies showed an association between the *ACE1* rs1799752 Ins allele and protection against the most severe form of COVID-19, in all inheritance models (OR = 0.67, 95% CI 0.56–0.82, **Table 2** and **Fig 2A** for the allele model). Hubacek *et al.* [47] and Cafiero *et al.* [32] studies only compared asymptomatic *vs.*

**Table 2. Meta-analyses of the association between polymorphisms in candidate genes and COVID-19 development and severity.**

| Polymorphism | Localization/Position | Inheritance model | Studies | I² | Model | OR (95% CI) |
|---|---|---|---|---|---|---|
| **COVID-19 infection *vs*. Control** | | | | | | |
| *ACE2* rs2285666 | chrX:15592225 / Intron | Dominant | 3 | 64.1% | Random | 0.95 (0.57–1.56) |
| *ACE2* rs41303171 | chrX:15564175 / Exon | Allele | 3 | 66.3% | Random | 1.52 (0.24–9.61) |
| | | Dominant | 3 | 67.8% | Random | 1.36 (0.20–9.20) |
| *ACE1 Ins/Del* | chr17:63488530–63488543 / Intron | Allele | 4 | 61.7% | Random | 1.00 (0.82–1.22) |
| | | Dominant | 4 | 64.1% | Random | 0.95 (0.70–1.28) |
| | | Recessive | 4 | 64.2% | Random | 0.93 (0.64–1.37) |
| | | Additive | 4 | 72.3% | Random | 0.89 (0.55–1.46) |
| *TMPRSS2* rs12329760 | chr21:41480570 / Exon | Allele | 3 | 12.6% | Fixed | 1.08 (0.92–1.27) |
| | | Dominant | 3 | 0% | Fixed | 1.18 (0.96–1.45) |
| *CCR5* rs333 | chr3:46373453–46373487 / Exon | Allele | 3 | 44.6% | Fixed | 0.80 (0.68–0.96)* |
| | | Dominant | 3 | 40.3% | Fixed | 0.82 (0.68–0.98)* |
| *ApoE* ε4 | chr19:44908684 and chr19:44908822† / Exon | Allele | 3 | 41.8% | Fixed | 1.32 (1.20–1.45)* |
| | | Dominant | 3 | 58.2% | Random | 1.38 (1.09–1.75)* |
| | | Recessive | 3 | 28.2% | Fixed | 1.94 (1.50–2.50)* |
| | | Additive | 3 | 27.1% | Fixed | 2.05 (1.58–2.65)* |
| *ABO* rs8176719 | chr9:133257521–133257522 / Exon | Allele | 3 | 80.7% | Random | 1.22 (0.99–1.49) |
| **COVID-19 mild/moderate *vs*. severe** | | | | | | |
| *ACE1Ins/Del* | chr17:63488530–63488543 / Intron | Allele | 5 | 45.4% | Fixed | 0.67 (0.56–0.82)* |
| | | Dominant | 5 | 41.4% | Fixed | 0.62 (0.47–0.83)* |
| | | Recessive | 5 | 0% | Fixed | 0.69 (0.50–0.95)* |
| | | Additive | 5 | 0% | Fixed | 0.49 (0.33–0.72)* |
| *TMPRSS2* rs12329760 | chr21:41480570 / Exon | Allele | 5 | 0% | Fixed | 0.77 (0.66–0.91)* |
| | | Dominant | 5 | 0% | Fixed | 0.74 (0.61–0.90)* |
| | | Recessive | 5 | 0% | Fixed | 0.71 (0.44–1.15) |
| | | Additive | 5 | 0% | Fixed | 0.65 (0.40–1.06) |
| *CCR5* rs333 | chr3:46373453–46373487 / Exon | Allele | 3 | 67.2% | Random | 0.83 (0.59–1.16) |
| | | Dominant | 3 | 67.4% | Random | 0.83 (0.58–1.18) |
| *IFITM3* rs12252 | chr11:320772 / Exon | Allele | 4 | 65.6% | Random | 1.04 (0.62–1.75) |
| | | Dominant | 4 | 65.8% | Random | 0.97 (0.53–1.77) |
| | | Recessive | 4 | 22.5% | Fixed | 1.04 (0.44–2.46) |
| | | Additive | 4 | 34.3% | Fixed | 0.78 (0.31–1.91) |
| *ApoE* ε4 | chr19:44908684 and chr19:44908822† / Exon | Allele | 3 | 0% | Fixed | 1.36 (1.07–1.73)* |
| | | Dominant | 3 | 0% | Fixed | 1.30 (0.97–1.72) |
| *ABO* rs8176719 | chr9:133257521–133257522 / Exon | Allele | 4 | 0% | Fixed | 0.94 (0.85–1.05) |

OR: odds ratio; CI: confidence interval.

* Indicates a significant association at P <0.05.

† Location of the two polymorphisms (rs429358 and rs7412) that generated the *ApoE* ε4 haplotype.

symptomatic patients, while the study be Calabrese *et al*. [33] compared groups according to the presence of thromboembolism in patients with severe COVID-19. Of note, when we included all the 8 studies in the meta-analysis, the Ins allele remained associated with protection against severe COVID-19 (OR = 0.60, 95% CI 0.39–0.94, for the allele model).

The *TMPRSS2* rs12329760 (C/T) polymorphism was analyzed in 3 studies regarding COVID-19 infection [52, 68, 73, 77] and 5 studies investigating disease severity [61, 68, 73, 83] (**Table 2**). Although the rs12329760 polymorphism was not associated with the risk of

(A)

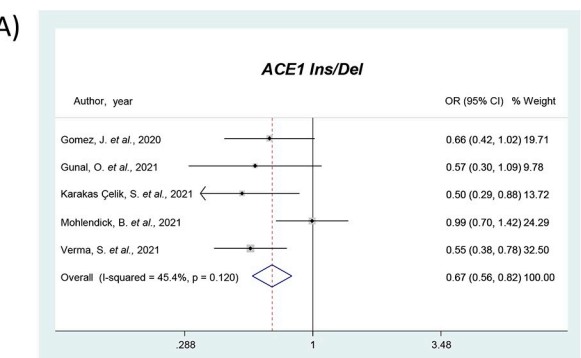

(B)

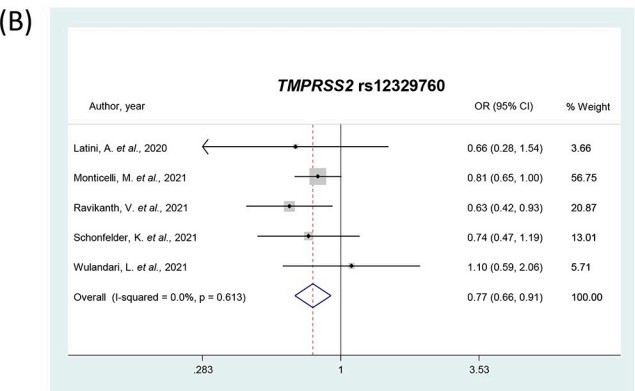

(C)

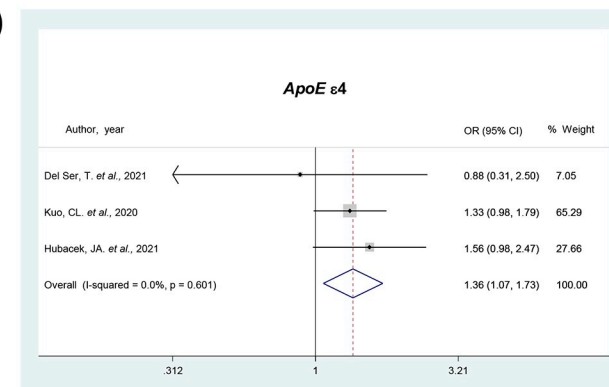

**Fig 2.** Forest plots showing individual and pooled ORs (95% CIs) for the associations between the *ACE1 Ins/Del* (**A**), *TMPRSS2* rs12329760 (**B**), and *ApoE* ε4 (**C**) polymorphisms and COVID-19 severity, under the allele contrast model.

COVID-19, this meta-analysis showed that the T allele of this polymorphism confers protection for the most severe form of COVID-19 when considering both allele (OR = 0.77, 95% CI 0.66–0.91; **Fig 2B**) and dominant model (OR = 0.74, 95% CI 0.61–0.90) models (**Table 2**).

## Meta-analyses of *HLA* alleles

The *A*, *B*, *C*, *DRB1*, *DQB1*, and *DQA1* alleles of the *HLA* were analyzed according to the risk of COVID-19 (**S3 Table**) or the severity of the disease (**S4 Table**). The *HLA-A\*30* allele was analyzed in 3 studies [39, 55, 56], and the pooled analysis showed this allele confers protection against COVID-19 (OR = 0.79, 95% CI 0.64–0.98; **S3 Table** and **Fig 3A**).

Regarding COVID-19 severity, the pooled data of 4 articles (5 studies) [39, 55, 56, 62] showed the association between the *HLA-A\*33* allele and protection for the most severe form of disease (OR = 0.56, 95% CI 0.36–0.88; **S4 Table** and **Fig 3B**). In contrast, the *HLA-B\*38* and *HLA-C\*06* alleles, both analyzed in the same 4 articles (5 studies) [39, 55, 56, 62], were associated with risk for the most severe form of COVID-19 (OR = 1.64, 95% CI 1.03–2.60 and OR = 1.31, 95% CI 1.00–1.72, respectively; **S4 Table** and **Fig 3C and 3D**). Our meta-analyses demonstrated that the other 70 alleles of the *A*, *B*, *C*, *DRB1*, *DQB1*, and *DQA1* loci were not associated with COVID-19 development or severity (**S3** and **S4 Tables**).

## Meta-analyses of *CCR5* and *IFITM3* polymorphisms

Three studies were included in the meta-analyses of *CCR5* rs333 (Ins/Del) polymorphism regarding the risk of COVID-19 and its severity [30, 36, 46] (**Table 2**). The Del allele was

(A)

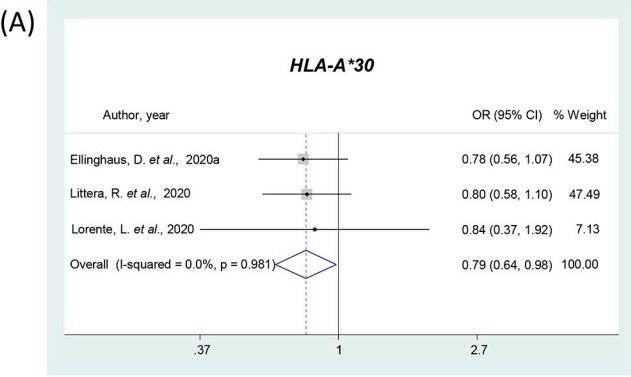

(B)

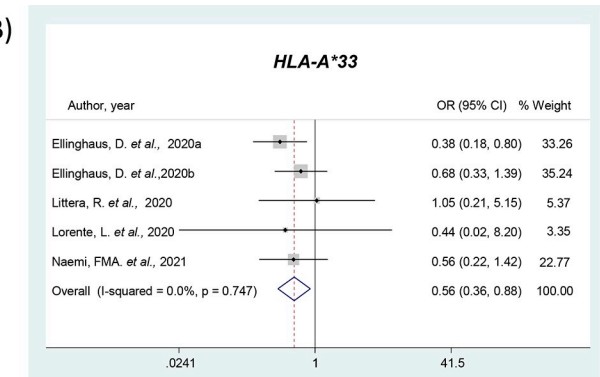

(C)

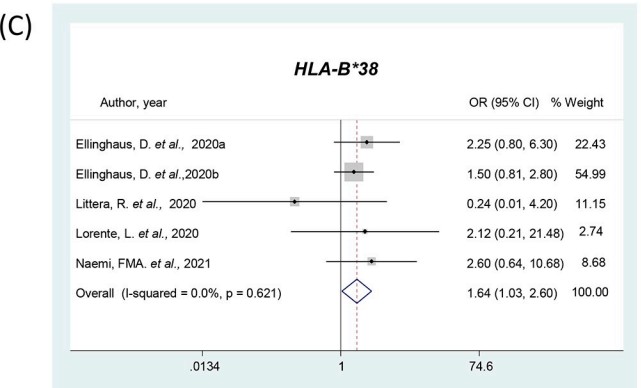

(D)

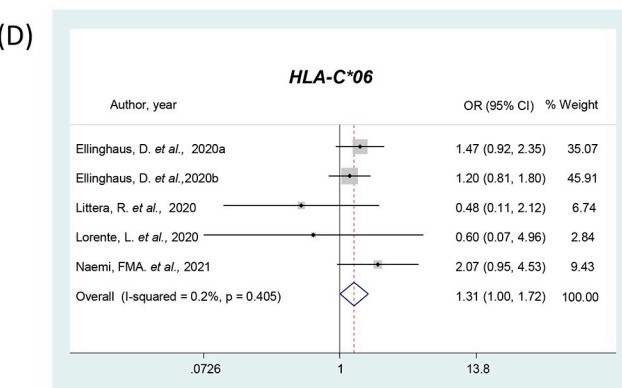

**Fig 3. Forest plots showing individual and pooled ORs (95% CIs) for the associations between *HLA* alleles and COVID-19 presence or severity. (A)** Forest plot for *HLA-A*30* and COVID-19 presence. **(B)** Forest plot for *HLA-A*33* and COVID-19 severity. **(C)** Forest plot for *HLA-B*38* and COVID-19 severity. **(D)** Forest plot for *HLA-B*06* and COVID-19 severity. [a] Data from an Italian population; [b] Data from a Spanish population.

associated with protection for COVID-19 infection considering both allele (OR = 0.80, 95% CI 0.68–0.96; **Fig 4A**) and dominant (OR = 0.82, 95% CI 0.68–0.98) models; however, this polymorphism was not associated with the severity of the disease (**Table 2**).

For the *IFITM3* rs12252 (T/C) polymorphism, the pooled analyses of 4 studies [24, 42, 72, 84] indicated no association of this polymorphism and different degrees of COVID-19 severity, for all tested genetic models (**Table 2**).

(A)

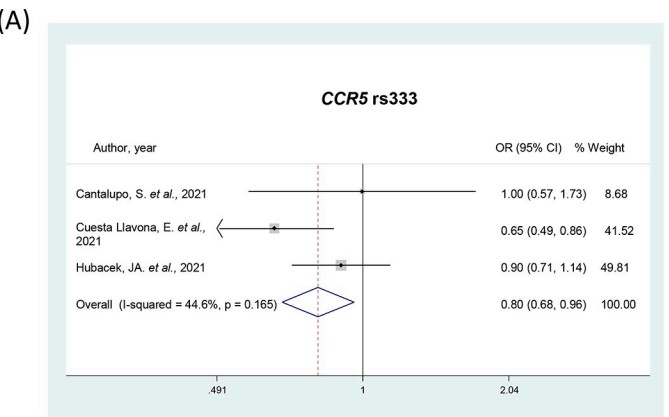

(B)

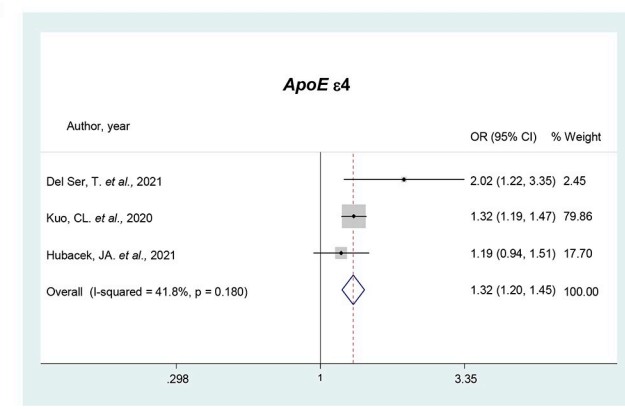

**Fig 4.** Forest plots showing individual and pooled ORs (95% CIs) for the associations between the *CCR5* rs333 (**A**) and *ApoE* ε4 (**B**) polymorphisms and COVID-19 presence, both under the allele contrast model.

## Meta-analyses of *ApoE* and *ABO* polymorphisms

The *ApoE* ε4 genotype was analyzed in 3 studies [37, 46, 51] regarding both COVID-19 infection and severity (**Table 2**). Meta-analyses showed the ε4 allele was associated with risk for COVID-19 presence in all genetic models (OR = 1.32, 95% CI 1.20–1.45, **Fig 4B** for the allele model). The ε4 allele was also associated with risk for the most severe form of COVID-19 when considering the allele model (OR = 1.36, 95% CI 1.07–1.73, **Fig 2C**).

The rs8176719 (-/C) polymorphism in the *ABO* gene was evaluated in 3 studies (2 articles) [38, 39] about COVID-19 development and 4 studies (3 articles) regarding disease severity [35, 38, 39] (**Table 2**). The pooled analyses indicated the Ins C allele is not associated with COVID-19 presence or severity in the allele model.

## Discussion

Elucidating the genetic determinants of SARS-CoV-2 infection is essential for understanding the pathophysiology of COVID-19 and the inter-individual variability in its severity; thus, contributing to the development of updated vaccines and new antivirals. Hence, in this systematic review, we summarized the results of 64 eligible articles that analyzed the association between genetic polymorphisms and risk for infection or severity of COVID-19. Moreover, data regarding polymorphisms in 8 genes (*HLA*, *ABO*, *ACE1*, *ACE2*, *APOE*, *CCR5*, *TMPRSS2*, and *IFITM3)* were meta-analyzed in relation to the risk of infection and severity of COVID-19. Pooled results demonstrated that polymorphisms in the *ApoE*, *ACE1*, *TMPRSS2*, *CCR5*, and *HLA* genes appear to be involved in the susceptibility to and/or severity of COVID-19.

Angiotensin-converting enzyme 2 (*ACE2*) and type II transmembrane serine protease (*TMPRSS2*) are candidate genes for susceptibility for SARS-CoV-2 infection since SARS-CoV-2 uses the ACE2 receptor for cell entry, while the serine protease TMPRSS2 is required for priming of the viral spike (S) protein [86, 87]. ACE2 and ACE1, together with renin and angiotensin, constitute the renin angiotensin aldosterone system (RAAS), which is a complex system involved in multiple biological process that regulated blood pressure homeostasis and extracellular volume, and inflammation, which is closely related to COVID-19 morbidity and mortality, as it affects bradykinin production [88, 89]. Following the viral entry, ACE2 is downregulated, causing an ACE1/ACE2 imbalance and contributing to RAAS overactivation and pulmonary shutdown. The consequent increased ACE1 activity and reduced ACE2 expression increase the risk of pulmonary diseases by increasing the lung vascular permeability; thus, leading to lung damage [90–92]. Accordingly, studies have reported the association between polymorphisms in *ACE1*, *ACE2*, and *TMPRSS2* genes and SARS-CoV-2 infection [28, 32, 33, 41, 44, 48, 52, 58, 60, 61, 63, 68, 73, 77, 83]; however, the results are still contradictory. In the present meta-analysis, two *ACE2* polymorphisms (rs2285666 and rs41303171) were analyzed, but no association with COVID-19 was found. Nevertheless, we demonstrated an association between the T allele of the *TMPRSS2* rs12329760 polymorphism and protection against the most severe form of COVID-19.

Regarding the *ACE1* gene, the insertion/deletion (Ins/Del) of 287-bp in the *Alu*-sequence of intron 16, represented by four individual SNPs (rs4646994, rs1799752, rs4340 and rs13447447), modulates *ACE1* expression [93–95]. This Ins/Del variant results in alternative splicing, leading to protein shortening and loss of the catalytically active domain in *ACE1* Ins allele carriers [92]. Moreover, the *ACE1* Ins/Del variant explains about 60% of variability in ACE1 levels in the general population since ACE1 levels in Ins/Ins carriers are approximately half of that of Del/Del carriers [39, 93, 96]. In the context of SARS-CoV-2 infection, studies have reported variations in COVID-19 recovery and prevalence rates are associated to *ACE1*

Ins/Del frequency and geographical variations of this variant [97, 98]. Here, we showed an association between the *ACE1* Ins allele and protection against severe COVID-19.

Major histocompatibility complex genes (*MHC*, known as Human Leukocyte Antigens, *HLA*) play a critical role in immune response [99]. The HLA system is a remarkably polymorphic region and genetic variants of *HLA* have been reported to affect the clinical course of patients infected with different viruses [100], including SARS-CoV-1 [101]. A specific set of HLA will present the peptides of the degraded virus to receptors on T cells, thus eliciting an immune response for virus eradication [102]. The set of *HLA* alleles inherited by an individual will determine the immune responses to viruses according to the selected peptides that can bind to the peptide-binding groove [102]. Studies in different populations have shown associations between some *HLA class I* (*A*, *B*, and *C*) and *class II* (*DRB1*, *DQA1*, and *DQB1*) alleles and COVID-19 susceptibility and/or severity [82, 103]. Our meta-analyses did not confirm the results of previous individual studies; however, we identified new *HLA* alleles associated with COVID-19: the *HLA-A*$^*$*30* and *HLA-A*$^*$*33* were associated with protection against COVID-19 infection and the most severe form of this disease, respectively. Besides, the *HLA-B*$^*$*38* and *HLA-C*$^*$*06* alleles were associated with risk for severe COVID-19.

The interferon-induced transmembrane 3 (IFITM3) is an IFN-stimulated gene (ISG) essentially expressed on endosomes and lysosomes [104]. IFITM3 is part of an ISG family (IFITM) responsible for inhibiting the fusion between viral and cellular membranes of many viruses, such as influenza A H1N1 virus, dengue virus, and SARS-CoV [104]. On the other hand, it was recently shown that IFITM proteins are cofactors for efficient SARS-CoV-2 infection in human cells [105], reaffirming a key role of this gene in the susceptibility to COVID-19. Nevertheless, here, the *IFITM3* rs12252 polymorphism was not associated with COVID-19 severity. Of note, we did not analyze this polymorphism regarding COVID-19 infection susceptibility due to lack of studies. Although this SNP in *IFITM3* gene was not associated with COVID-19, it is noteworthy that type I IFN (IFN-I)-stimulated immunity has been shown to influence COVID-19 severity. Inborn errors of IFN-I pathway and pre-existing autoantibodies neutralizing IFN-I appear to be strong determinants of critical COVID-19 pneumonia in about 15–20% of patients [106]. Asano *et al.*, [107] reported that deleterious X-linked *TLR7* mutations were observed in 16 male subjects from a cohort of 1202 patients with unexplained critical COVID-19 pneumonia. The patients' blood plasmacytoid dendritic cells (pDCs) produced low levels of IFN-I in response to SARS-CoV-2. Human TLR7 and pDCs are essential for protective IFN-I immunity against SARS-CoV-2 in the respiratory tract. Moreover, Zhang *et al.*, [108] showed that inborn errors of TLR3- and IRF-7 dependent IFN-I immunity can cause life-threatening COVID-19 pneumonia in patients with no prior severe infection.

Chemokines act attempting to maintain the immune homeostasis and to defend the body against harmful stimuli, such as SARS-CoV-2 infection [109]. *CCR5* encodes a chemokine receptor expressed in macrophages and T cells, and its upregulation has been confirmed in COVID-19 patients [110]. Furthermore, an anti-CCR5 treatment has been shown to relieve the symptoms and the cytokine storm in COVID-19 patients who are critically ill [109]. The *CCR5* gene is located at 3p21.31, a gene cluster region associated with severe COVID-19 courses [39]. The most studied *CCR5* polymorphism regarding COVID-19 susceptibility is the Δ32 Ins/Del (rs333) [30, 34, 36, 46]. The *CCR5* rs333 Del allele results in loss of function of the protein; being a major determinant of the resistance to HIV infection since the CCR5 protein serves as one of the gateways for the HIV virus [111]. Accordingly, our meta-analysis showed the *CCR5* rs333 Del allele was associated with protection against COVID-19 infection [34, 36, 46].

A Genome-Wide Association Study (GWAS) carried out by the Severe COVID-19 GWAS Group [39] reported that one of the 2 strongest signals associated with severe COVID-19 was located within the ABO blood-group system. The involvement of ABO blood groups in COVID-19 susceptibility has been reported in both genetic and non-genetic studies. The blood group O was previously associated with a lower risk of acquiring COVID-19 when compared to subjects with non-O blood groups, whereas the blood A group was associated with a higher risk for this disease than non-A blood groups [39]. One of the assumptions is that the A-antigen causes P-selectin and intercellular cell adhesion molecule 1 binding to endothelial cells, increasing the probability of cardiovascular disease. Another explanation is that individuals with blood group O have decreased levels of von Willebrand factor, lowering the thrombotic disease risk [reviewed in [103]]. The rs8176719 polymorphism is the main determinant of the O blood group and has been investigated as a potential marker of COVID-19 susceptibility. However, some studies did not confirm these findings [35, 38]. In our meta-analysis, we demonstrated that the *ABO* rs8176719 - /C SNP was not associated with COVID-19 infection neither with different stages of severity.

The *ApoE* ε4 genotype was investigated in the UK Biobank Cohort, being associated with COVID-19 severity and mortality [51]. This finding was replicated in other studies [37, 46]. Apolipoprotein E (ApoE) is broadly expressed in human tissues and has an essential role in lipid transport, which has a key role in many functions, including immunity [112]. The most studied polymorphisms in *ApoE* are the rs429358 (ApoE4, C/T) and rs7412 (ApoE2, C/T), both located at exon 4. Three haplotypes are generated from these two polymorphisms (ε2, ε3 and ε4), codifying 3 protein isoforms (E2, E3 and E4). Moreover, these haplotypes can combine in 6 different variants: ε2/ε2, ε2/ε3, ε2/ε4, ε3/ε3, ε3/ε4, and ε4/ε4 [112]. Among them, the ancestral *ApoE* ε4/ε4, generally considered deleterious, is a significant risk factor for Alzheimer's disease and other human pathologies, including type 2 diabetes and cardiovascular disease, which are known risk factors for worst outcomes of COVID-19 [112–114]. In the present meta-analysis, the pooled data of three studies confirmed the association of the ε4 allele with both risk to COVID-19 presence and severe outcomes of the disease. It has been hypothesized that elevated cholesterol and oxidized lipoprotein levels, linked to the effects of ApoE ε4/ε4 variant, is associated with increased pneumocyte susceptibility to infection and to exaggerated lung inflammation [112]. Moreover, the frequency of the ε4 allele is higher in African-Americans who had increased mortality due to COVID-19 compared to Caucasian populations [115].

The results of the present meta-analysis should be interpreted within the context of a few limitations. Inter-studies heterogeneity is common in meta-analyses of genetic association studies and it should be cautiously interpreted. Some included studies did not test the control groups for COVID-19 or included controls derived from previous databank or ecological studies without COVID-19 information. Moreover, the COVID-19 severity criteria varied among the studies. Particular studies had included asymptomatic patients while others only included patients with at least a given symptom. Due to the presence of more than 2 groups of COVID-19 severity stages (mild, moderate and severe), we have categorized the patients regarding COVID-19 severity in different ways; however, it was more rational to show the data categorizing the most severe group against the others groups (asymptomatic and/or mild plus moderate). It was not possible to evaluate the association with mortality, as only few studies presented data comparing COVID-19 survivors and non-survivors. Furthermore, the impact of gender and age, which may influence the COVID-19 predisposition, could not be assessed due to the small number of studies for each SNP. Genetic background among different populations may significantly influence COVID-19 susceptibility, and the studies included in the present meta-analysis comprised different ethnicities. However, due to the small number of

studies for each ethnicity, we were not able to analyze the impact of genetic background on the results. Finally, we cannot be sure that small negative studies were overlooked since we could not perform the publication bias analysis due to the small amount of studies for each SNP.

The infection with SARS-CoV-2 and its clinical course are dependent on the complex relationship between the virus and the host immune system. In this meta-analysis, we identified, for the first time, that four alleles of the *HLA class I* loci (*A*30*, *A*33*, *B*38* and *C*06*) are associated with COVID-19. Moreover, we confirmed the association between COVID-19 susceptibility and polymorphisms in the *ApoE*, *ACE1*, *TMPRSS2*, and *CCR5* genes. These findings will guide further epidemiological studies on host genetics as well as the development of innovative treatments. Considering that specific genetic polymorphisms might lead to severe COVID-19 outcomes, it is of extreme importance to use individual genetic data to employ personalized therapeutics and improve the COVID-19 prognostic.

## Supporting information

**S1 Table. Clark-Baudouin quality assessment scale for the studies included in the systematic-review.**
(DOCX)

**S2 Table. Characteristics of studies included in this systematic review and meta-analysis.**
(XLSX)

**S3 Table. Meta-analyses of the association between polymorphisms in HLA and COVID-19.**
(DOCX)

**S4 Table. Meta-analyses of the association between polymorphisms in *HLA* and COVID-19 severity.**
(DOCX)

## Author Contributions

**Conceptualization:** Cristine Dieter, Letícia de Almeida Brondani, Fernando Gerchman, Natália Emerim Lemos, Daisy Crispim.

**Data curation:** Cristine Dieter, Letícia de Almeida Brondani, Natália Emerim Lemos.

**Formal analysis:** Cristine Dieter, Letícia de Almeida Brondani, Daisy Crispim.

**Funding acquisition:** Cristiane Bauermann Leitão, Fernando Gerchman.

**Investigation:** Cristine Dieter, Letícia de Almeida Brondani, Daisy Crispim.

**Methodology:** Cristine Dieter, Letícia de Almeida Brondani, Natália Emerim Lemos.

**Project administration:** Cristiane Bauermann Leitão, Fernando Gerchman, Daisy Crispim.

**Resources:** Cristine Dieter, Letícia de Almeida Brondani, Daisy Crispim.

**Software:** Cristine Dieter, Letícia de Almeida Brondani.

**Supervision:** Daisy Crispim.

**Writing – original draft:** Cristine Dieter, Letícia de Almeida Brondani, Cristiane Bauermann Leitão, Fernando Gerchman, Natália Emerim Lemos, Daisy Crispim.

**Writing – review & editing:** Cristine Dieter, Letícia de Almeida Brondani, Cristiane Bauermann Leitão, Fernando Gerchman, Natália Emerim Lemos, Daisy Crispim.

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
