## [Decision Letter · Decision Letter 0]

18 Apr 2022

PONE-D-22-06959Genetic polymorphisms associated with susceptibility to COVID-19: a systematic review and meta-analysisPLOS ONE

Dear Dr. Crispim,

Thank you for submitting your manuscript to PLOS ONE. After careful consideration, we feel that it has merit but does not fully meet PLOS ONE’s publication criteria as it currently stands. Therefore, we invite you to submit a revised version of the manuscript that addresses the points raised during the review process.

Please integrate in the new version of the manuscript with the suggestions of the reviewers and myself. In particular, to have a greater impact success, it would be advisable for the authors to dwell more on the concrete results obtained from the study of candidate genes such as those of the interferon circuit (SEE EDITOR'S COMMENT) This notion is supported by an extensive sequencing of numerous patients with severe forms of COVID-19 who have identified pathogenic mutations in genes that code for active proteins in the interferon circuit. The characterization of autoantibodies capable of neutralizing IFN-I in 10-15% of severe patients allows us to state that COVID-19 can be defined as an interferonopathy. This must be included in Discussion.

We look forward to receiving your revised manuscript.

Kind regards,

Giuseppe Novelli

Academic Editor

PLOS ONE

Journal Requirements:

"This study was partially supported by grants from the Conselho Nacional de

Desenvolvimento Científico e Tecnológico (CNPq, grant number 401610/2020-9),

Fundo de Incentivo à Pesquisa e Eventos (FIPE) at Hospital de Clínicas de Porto Alegre

(grant number: 2020-0218), and Coordenação de Aperfeiçoamento de Pessoal de Nível

Superior (CAPES). D.C., C.B.L. and N.E.L are recepients of a scholarship from CNPq,

while C.D. is a recipient of scholarship from CAPES. "

"This study was partially supported by grants from the Conselho Nacional de Desenvolvimento Científico e Tecnológico (CNPq, grant number 401610/2020-9), Fundo de Incentivo à Pesquisa e Eventos (FIPE) at Hospital de Clínicas de Porto Alegre (grant number: 2020-0218), and Coordenação de Aperfeiçoamento de Pessoal de Nível Superior (CAPES). D.C., C.B.L. and N.E.L are recepients  of a scholarship from CNPq, while C.D. is a recipient of scholarship from CAPES. "

Additional Editor Comments:

A comprehensive study should include some of the most notable findings from the past two years in this field. Indeed, several clinical and immunological studies have shown that type I interferons (IFN-I) play critical roles in the control and pathogenesis of COVID-19. This notion is supported by extensive sequencing of numerous patients with severe COVID-19 who have identified pathogenic mutations in genes that code for active proteins in the interferon circuit. The characterization of autoantibodies capable of neutralizing IFN-I in 10-15% of severe patients allows us to state that COVID-19 can be defined as an interferonopathy. This must be included in Discussion and the references below must be cited:

Zhang Q, Bastard P, Effort CHG, Cobat A, Casanova JL. Human genetic and immunological determinants of critical COVID-19 pneumonia. Nature. 2022. https://doi.org/10.1038/s41586-022-04447-0. Epub ahead of print.

Asano T, Boisson B, Onodi F, Matuozzo D, Moncada-Velez M, Maglorius Renkilaraj MRL, et al. X-linked recessive TLR7 deficiency in ~1% of men under 60 years old with life-threatening COVID-19. Sci Immunol. 2021 Aug 19;6(62):eabl4348. https://doi.org/10.1126/sciimmunol.abl4348

Zhang Q, Bastard P, Liu Z, Le Pen J, Moncada-Velez M, Chen J, et al. Inborn errors of type I IFN immunity in patients with life-threatening COVID-19. Science. 2020 Oct 23;370(6515):eabd4570. https://doi.org/10.1126/science.abd4570. Epub 2020 Sep 24.

Reviewers' comments:

Reviewer's Responses to Questions

**Comments to the Author**

1. Is the manuscript technically sound, and do the data support the conclusions?

Reviewer #1: Yes

Reviewer #2: Yes

2. Has the statistical analysis been performed appropriately and rigorously? 

Reviewer #1: Yes

Reviewer #2: Yes

3. Have the authors made all data underlying the findings in their manuscript fully available?

Reviewer #1: Yes

Reviewer #2: Yes

4. Is the manuscript presented in an intelligible fashion and written in standard English?

Reviewer #1: Yes

Reviewer #2: Yes

5. Review Comments to the Author

Reviewer #1: The manuscript describes a systematic review of the literature regarding the possible involvement of genetic factors in the susceptibility to SARS-CoV-2 infection.

The main points are consistent with the analysis carried out, according to the Preferred Reporting Items for Systematic Review and Meta-analysis statement (PRISMA). The inclusion and exclusion criteria are well reported. The authors provide a needed synopsis on the current status of the topic, shedding light on the limitations and future perspectives that might be useful for future analysis. The structure of the review is clear and well organised.

The article is interesting and well-focused in the Methods, Results and Discussion parts.

I have only few comments:

Table 1 shows all the studies included in the systematic review. They would be easier to consult if they were split between investigating genetic factors that may influence COVID-19 susceptibility and those involved in severity.

Moreover, P-value and OR are should be reported, where possible, for all included studies.

The location of the polymorphisms included in meta-analyses should be indicate.

The title of the paragraph " Search strategy and eligibility criteria" could also be “Literature Search strategy and eligibility criteria”

I suggest to further discuss the issue of genetic variability among different populations, since the majority of studies were conducted in different ethnic groups.

Reviewer #2: In this manuscript the authors provide a systematic review and meta-analysis of current literature, investigating the association of polymorphisms with COVID-19 susceptibility and severity. The objectives of the analysis are clearly stated and the informations on the search are provided (sources, used terms for literature search). Inclusion and exclusion criteria are clearly stated. Characteristics of the selected studies are complete and well resumed in Table 1 and S2 table. Statistical methods seems appropriate and results are well displayed. I particularly appreciate that the authors have correctly probed the limitations of this study, which can’t be ignored when interpreting the results.

In the manuscript, there are only few minor flaws to be addressed:

1) the title should emphasize the analyses conducted on the association between genetic factors and COVID-19 severity too, since in the discussion it has an equal relevance compared to analyses on the association with COVID-19 susceptibility;

2) reference for the Clark-Baudouin Score (“Data extraction and quality evaluation” section) should be checked, since it doesn’t seem correct;

3) I suggest to rephrase the statement “Different comorbidities are associated with a worse COVID-19 outcome, and dementia was among the common comorbidities linked with higher mortality” in the discussion section, since it could convey a message not yet fully supported by scientific evidence, although I can see that is not in the authors’ intentions. As far as I know, there are no studies that have been able to significantly discriminate the contribution of different factors that may underlie an increased risk of COVID-19 mortality in patients with dementia. Due to the characteristic of the pathology, it is not possible not to recognize the relevance of socioeconomic and behavioral factors (failure to observe preventive measures or adherence to therapy). It must also be taken into account that some conditions predisposing to dementia are also risk factors for adverse COVID-19 outcomes (cardiovascular diseases, type 2 diabetes, obesity, asthma, chronic kidney disease). So there are still deeper investigations to be done before dementia itself can be listed as a risk factor associated with higher COVID-19 mortality. Moreover, the cited paper for this statement stresses the focus on the neurological complications of a SARS-CoV-2 infection;

4) unfortunately, it’s not possible to state that this is the first meta-analysis on the field (PMID:34997794), but in my opinion this doesn’t affect the validity of this work, since it is not focused only on the genetic polymorphisms in genes related to the renin-angiotensin-aldosterone system (RAAS), as the cited one.

In conclusion, for what is in my competence, the manuscript seems carefully conceived and well written. This work, on its current form, provides a promising starting point and it can have an impact in terms of designing broader and deeper investigations.

6. PLOS authors have the option to publish the peer review history of their article (what does this mean?). If published, this will include your full peer review and any attached files.

Reviewer #1: No

Reviewer #2: No

---

## [Author Response · Author response to Decision Letter 0]

22 Apr 2022

To

Giuseppe Novelli

Dear Editor:

Thank you for your letter concerning our manuscript. 

We appreciated the Reviewer's comments, which helped us to further improve the manuscript. In the following pages you will find our answers for each of the Reviewer's comments. We hope that you will find these answers and the revised version of the manuscript satisfactory. All modifications to the original manuscript are indicated by red fountain.

We look forward to receive your feedback on this revised manuscript.

Sincerely yours,

Profª. Drª. Daisy Crispim

Point-by-point answers to the Editor comments:

A comprehensive study should include some of the most notable findings from the past two years in this field. Indeed, several clinical and immunological studies have shown that type I interferons (IFN-I) play critical roles in the control and pathogenesis of COVID-19. This notion is supported by extensive sequencing of numerous patients with severe COVID-19 who have identified pathogenic mutations in genes that code for active proteins in the interferon circuit. The characterization of autoantibodies capable of neutralizing IFN-I in 10-15% of severe patients allows us to state that COVID-19 can be defined as an interferonopathy. This must be included in Discussion and the references below must be cited:

Zhang Q, Bastard P, Effort CHG, Cobat A, Casanova JL. Human genetic and immunological determinants of critical COVID-19 pneumonia. Nature. 2022. https://doi.org/10.1038/s41586-022-04447-0. Epub ahead of print.

Asano T, Boisson B, Onodi F, Matuozzo D, Moncada-Velez M, Maglorius Renkilaraj MRL, et al. X-linked recessive TLR7 deficiency in ~1% of men under 60 years old with life-threatening COVID-19. Sci Immunol. 2021 Aug 19;6(62):eabl4348. https://doi.org/10.1126/sciimmunol.abl4348

Zhang Q, Bastard P, Liu Z, Le Pen J, Moncada-Velez M, Chen J, et al. Inborn errors of type I IFN immunity in patients with life-threatening COVID-19. Science. 2020 Oct 23;370(6515):eabd4570. https://doi.org/10.1126/science.abd4570. Epub 2020 Sep 24.

Answer: Thank you for the suggestions. We included a discussion about this topic on the Discussion Section (page 34).

Point-by-point answers to the Reviewer’s comments:

Reviewer #1: 

The manuscript describes a systematic review of the literature regarding the possible involvement of genetic factors in the susceptibility to SARS-CoV-2 infection.

The main points are consistent with the analysis carried out, according to the Preferred Reporting Items for Systematic Review and Meta-analysis statement (PRISMA). The inclusion and exclusion criteria are well reported. The authors provide a needed synopsis on the current status of the topic, shedding light on the limitations and future perspectives that might be useful for future analysis. The structure of the review is clear and well organised.

The article is interesting and well-focused in the Methods, Results and Discussion parts.

Answer: Thank you for your comments.

I have only few comments:

Table 1 shows all the studies included in the systematic review. They would be easier to consult if they were split between investigating genetic factors that may influence COVID-19 susceptibility and those involved in severity.

Moreover, P-value and OR are should be reported, where possible, for all included studies.

Answer: Thank you for your suggestion. We now included the P-values and ORs, when available, for all included studies. Since many studies reported both results for COVID-19 susceptibility and severity, we opted to not split the table according to results. However, we re-organized our table to make easy to identify those studies associated with susceptibility, severity or both.

The location of the polymorphisms included in meta-analyses should be indicate.

Answer: Thank you for your suggestion. We added the information about the location of the polymorphisms included in meta-analyses on Table 2 (page 24).

The title of the paragraph " Search strategy and eligibility criteria" could also be “Literature Search strategy and eligibility criteria”

Answer: We have changed the title according to your suggestion (page 4).

I suggest to further discuss the issue of genetic variability among different populations, since the majority of studies were conducted in different ethnic groups.

Answer: Thank you for the suggestion. We now included a discussion about genetic variability on the limitation paragraph of the Discussion Section (page 37).

Reviewer #2: 

In this manuscript the authors provide a systematic review and meta-analysis of current literature, investigating the association of polymorphisms with COVID-19 susceptibility and severity. The objectives of the analysis are clearly stated and the informations on the search are provided (sources, used terms for literature search). Inclusion and exclusion criteria are clearly stated. Characteristics of the selected studies are complete and well resumed in Table 1 and S2 table. Statistical methods seems appropriate and results are well displayed. I particularly appreciate that the authors have correctly probed the limitations of this study, which can’t be ignored when interpreting the results.

Answer: Thank you for your comment.

In the manuscript, there are only few minor flaws to be addressed:

1) the title should emphasize the analyses conducted on the association between genetic factors and COVID-19 severity too, since in the discussion it has an equal relevance compared to analyses on the association with COVID-19 susceptibility;

Answer: Thank you for your suggestion. We now also emphasized on the title the association with COVID-19 severity. 

2) reference for the Clark-Baudouin Score (“Data extraction and quality evaluation” section) should be checked, since it doesn’t seem correct;

Answer: We have now included the correct reference (page 6).

3) I suggest to rephrase the statement “Different comorbidities are associated with a worse COVID-19 outcome, and dementia was among the common comorbidities linked with higher mortality” in the discussion section, since it could convey a message not yet fully supported by scientific evidence, although I can see that is not in the authors’ intentions. As far as I know, there are no studies that have been able to significantly discriminate the contribution of different factors that may underlie an increased risk of COVID-19 mortality in patients with dementia. Due to the characteristic of the pathology, it is not possible not to recognize the relevance of socioeconomic and behavioral factors (failure to observe preventive measures or adherence to therapy). It must also be taken into account that some conditions predisposing to dementia are also risk factors for adverse COVID-19 outcomes (cardiovascular diseases, type 2 diabetes, obesity, asthma, chronic kidney disease). So there are still deeper investigations to be done before dementia itself can be listed as a risk factor associated with higher COVID-19 mortality. Moreover, the cited paper for this statement stresses the focus on the neurological complications of a SARS-CoV-2 infection;

Answer: As suggested, we rephrased this paragraph (Discussion Section, page 35).

4) unfortunately, it’s not possible to state that this is the first meta-analysis on the field (PMID:34997794), but in my opinion this doesn’t affect the validity of this work, since it is not focused only on the genetic polymorphisms in genes related to the renin-angiotensin-aldosterone system (RAAS), as the cited one.

Answer: Thank you for the information. We excluded this sentence from the Introduction Section.

In conclusion, for what is in my competence, the manuscript seems carefully conceived and well written. This work, on its current form, provides a promising starting point and it can have an impact in terms of designing broader and deeper investigations.

Answer: Thank you for your comment.

---

## [Editor Report · Decision Letter 1]

15 Jun 2022

Genetic polymorphisms associated with susceptibility to COVID-19 disease and severity: a systematic review and meta-analysis

PONE-D-22-06959R1

Dear Dr. Crispim,

We’re pleased to inform you that your manuscript has been judged scientifically suitable for publication and will be formally accepted for publication once it meets all outstanding technical requirements.

Kind regards,

Giuseppe Novelli

Academic Editor

PLOS ONE
---

## [Editor Report · Acceptance letter]

20 Jun 2022

PONE-D-22-06959R1 

Genetic polymorphisms associated with susceptibility to COVID-19 disease and severity: a systematic review and meta-analysis 

Dear Dr. Crispim:

I'm pleased to inform you that your manuscript has been deemed suitable for publication in PLOS ONE. Congratulations! Your manuscript is now with our production department. 

Kind regards, 

on behalf of

Prof. Giuseppe Novelli 

Academic Editor

PLOS ONE